

# Characterization of vaginal microbiota in Thai women

Auttawit Sirichoat[1], Pranom Buppasiri[2], Chulapan Engchanil[1], Wises Namwat[1], Kiatichai Faksri[1], Nipaporn Sankuntaw[3], Ekawat Pasomsub[4], Wasun Chantratita[5] and Viraphong Lulitanond[1]

[1] Department of Microbiology and Research and Diagnostic Center for Emerging Infectious Diseases, Faculty of Medicine, Khon Kaen University, Khon Kaen, Thailand
[2] Department of Obstetrics and Gynecology, Faculty of Medicine, Khon Kaen University, Khon Kaen, Thailand
[3] Chulabhorn International College of Medicine, Thammasat University, Pathum Thani, Thailand
[4] Division of Virology, Department of Pathology, Faculty of Medicine Ramathibodi Hospital, Mahidol University, Bangkok, Thailand
[5] Medical Genome Center, Faculty of Medicine, Ramathibodi Hospital, Mahidol University, Bangkok, Thailand

## ABSTRACT

**Background**. The vaginal microbiota (VMB) plays a key role in women's reproductive health. VMB composition varies with ethnicity, making it necessary to characterize the VMB of the target population before interventions to maintain and/or improve the vaginal health are undertaken. Information on the VMB of Thai women is currently unavailable. We therefore characterized the VMB in normal Thai women.
**Methods**. Vaginal samples derived from 25 Thai women were subjected to 16S rRNA gene next-generation sequencing (NGS) on the Ion Torrent PGM platform.
**Results**. Two groups of VMB were detected, lactobacilli-dominated (LD) and non-lactobacilli dominated (NLD) groups. *Lactobacillus iners* was the most common species found in the LD group while *Gardnerella vaginalis* followed by *Atopobium vaginae* and *Pseudumonas stutzeri* were commonly found in the NLD group.
**Conclusions**. The VMB patterns present in normal Thai women is essential information to further determine the factors associated with VMB patterns in vaginal health and disease and to develop proper management of reproductive health of Thai women.

## INTRODUCTION

The vaginal microbiota (VMB) plays an important role in vaginal health and in particular it has a protective role against vaginal infection. The composition of the VMB varies greatly among ethnic groups (*Ravel et al., 2011*). Up to now there was no definite study about how the VMB varies by ethnicity; however, some studies suggested the differences in VMB by ethnicities may be related to host genetic variations, which is associated with individual differences in immune response (*Blekhman et al., 2015*; *Green, Zarek & Catherino, 2015*; *Ma et al., 2014*). In addition, VMB composition can be affected by various internal and external factors, e.g., host physiology, transitional reproductive period, menstruation, pregnancy, infections, birth control procedures and amount of sexual activity

Corresponding author
Viraphong Lulitanond,
viraphng@kku.ac.th,
viraphng@gmail.com

(*Eschenbach et al., 2000*; *Gajer et al., 2012*; *Huang et al., 2014*; *Romero et al., 2014*; *Smith & Ravel, 2017*; *Vodstrcil et al., 2017*). In reproductive-aged women, the VMB is dominated by various *Lactobacillus* species such as *L. iners*, *L. crispatus*, *L. gasseri*, and *L. jensenii* (*Gajer et al., 2012*; *Ravel et al., 2011*). *In vitro* studies found that Lactobacilli can inhibit pathogens through the production of hydrogen peroxide and lactic acid which could maintain an acidic environment in the vagina (*Aroutcheva et al., 2001*; *Eschenbach et al., 1989*).

Several studies have characterized the VMB in healthy, reproductive women of different ethnicities using DNA sequence-based methods, especially next-generation sequencing (NGS) techniques (*Borgdorff et al., 2017*; *Fettweis et al., 2014*; *Ravel et al., 2011*). Five community state types (CSTs) have been recognized based on bacterial relative abundance on the basis of 16S ribosomal RNA (rRNA) sequences obtained using NGS (*Ravel et al., 2011*; *Zhou et al., 2004*). Four of these CSTs, i.e., CST-I, CST-II, CST-III and CST-V, were dominated by *L. crispatus*, *L. gasseri*, *L. iners* and *L. jensenii*, respectively. An additional type, CST-IV, is comprised of a high diversity of strict and facultative anaerobic bacteria including members of the genera *Gardnerella*, *Atopobium*, *Corynebacterium*, *Prevotella*, *Mobiluncus*, *Anaerococcus*, *Sneathia* and others, which are associated with vaginal infections especially bacterial vaginosis (BV) (*Fredricks, Fiedler & Marrazzo, 2005*; *Ling et al., 2010*; *Ravel et al., 2011*; *Swidsinski et al., 2005*). NGS-based studies have previously explored the VMB in Asian, African and European women (*Borgdorff et al., 2017*; *Chaban et al., 2014*; *Fettweis et al., 2014*; *Ravel et al., 2011*; *Van de Wijgert et al., 2014*). Among Asian populations, VMB in Chinese (*Ling et al., 2013*; *Shi et al., 2009*), Japanese (*Yoshimura et al., 2011*; *Zhou et al., 2010*) and South Korean (*Hong et al., 2016*; *Lee et al., 2013*) populations have already been studied, but women in South East Asia, including Thailand, have not been investigated. To our knowledge, this is the first report to characterize the VMB in normal Thai women based on 16S rRNA gene sequences obtained using NGS.

# MATERIAL AND METHODS

## Subject selection and sample collection

Twenty-five women attending the gynecologic out-patient clinics for cervical cancer screening, Srinagarind Hospital, Faculty of Medicine, Khon Kaen University, were enrolled in this study. The inclusion criteria used for subject selection specified non-pregnant women aged 20–45 years with regular menstruation, with normal vaginal examination and without any clinical symptoms on examination by one gynecologist. The exclusion criteria were women with the following conditions: having menstruation or sexual intercourse within the previous 24 h, use of antibiotics or vaginal antimicrobials within the last month, presence of any intravaginal product, use of a vaginal douche in the past week, and active infection or diagnosis of any defined sexually transmitted disease. The subjects selected by the above criteria were defined as normal women. All participants provided written informed consent and filled out a questionnaire detailing their level of education, employment, sociodemographic characteristics, dietary habits, reproductive health habits, and sexual behaviors (including number of children), income per month, alcohol consumption per week, exercise per week, sexual activity per week, number of sexual partners in the past,

regularity of their menstrual cycles in the last six months, current hormone contraceptive use, douching practice in the past month, abnormal vaginal discharge in the past month and female sterilization. This study was approved by the Ethics Committee of Khon Kaen University (HE601017).

A total of four points at mid right and left, anterior and posterior vaginal wall were swabbed with sterile cotton-tipped applicators by one gynecologist. The swabs were placed in sterile containers with phosphate-buffer saline (PBS) and stored at −80 °C until analysis.

## Genomic DNA extraction from vaginal swabs

Genomic DNA was extracted using the method described by *Ravel et al. (2011)* with slight modification. In briefly, the vaginal swabs were thawed on ice and adhered material resuspended by vigorous vortexing for 5 min. Bacterial cells were lysed by incubation for 1 h at 37 °C in TE50 buffer (10 mM Tris HCl, 50 mM EDTA, pH 8.0) containing 10 mg/mL of lysozyme (Merck, Darmstadt, Germany), 25,000 U/mL of mutanolysin (Sigma Aldrich, St. Louis, MO, USA) and 4,000 U/mL of lysostaphin (Sigma-Aldrich), followed by beating with 40–400 µm glass beads (NucleoSpin® Bead Tubes Type B; Macherey-Nagel, Düren, Germany) at speed 10 for 2 min with air cooling using the Bullet Blender® Blue Homogenizer (BBX 24B; Next Advance, Inc., Troy, NY, USA). After bead-beating, the crude lysates were processed for purification of genomic DNA using QIAamp® DNA Mini Kit (Qiagen, Hilden, Germany) according to the manufacturer's recommendations. Integrity, size and concentration of the purified genomic DNA was determined using Fragment Analyzer™ (Advanced Analytical Technologies, Inc., Ankeny, IA, USA).

## 16S rRNA gene sequencing by NGS

16S rRNA gene sequencing was done as previously described (*Barb et al., 2016*) with minor modifications. The seven 16S hypervariable regions were amplified with two primer sets, one set for amplifying the V2, V4, V8 hypervariable regions and the other set for amplifying the V3, V6-7, V9 hypervariable regions. Primers were supplied with the Ion 16S™ Metagenomics Kit (Life Technologies, Grand Island, NY, USA) and used according to the manufacturer's recommendations. The combined PCR products were processed to make the DNA library by using the Ion Plus™ Fragment Library Kit (Life Technologies, Carlsbad, CA, USA) with sample indexing using the Ion Xpress™ Barcodes Adapters 1-32 Kit (Life Technologies). The adapter-ligated and nick-repaired DNA was amplified with the following steps: 1 hold at 95 ° C for 5 min; 5–7 cycles of 95 °C for 15 s, 58 °C for 15 s, 70 °C for 1 min; hold at 4 °C. The PCR products were then purified using Agencourt® AMPure® XP Reagent (Beckman Coulter, Inc, Atlanta, GA, USA). The processed libraries were quantified using a Bioanalyzer® instrument (Agilent Technologies, Santa Clara, CA, USA) with the use of the Agilent® High-Sensitivity DNA Kit. Equal volumes of all 25 samples were pooled and processed with the Ion PGM™ Hi-Q™ OT2 Kit in the Ion OneTouch™ 2 System (Life Technologies) according to the manufacturer' instructions. Sequencing was performed in an Ion 318™ Chip (Life Technologies) using the Ion PGM™ Sequencing 400 Kit on the Ion PGM™ System (Life Technologies).

## Sequence data analysis

Base calling and run demultiplexing were performed using Torrent Suite™ Software version 5.0.5 (Life Technologies) with default parameters. All sequences were trimmed according to the quality. To pass, a sequence read had a following criteria: (1) a perfect match to a barcode sequence and the primer; (2) was at least 200 bp in length; (3) had no more than two undetermined bases; and (4) had an average quality score greater than Q20. Sequences were analyzed using Ion Reporter™ Software version 5.6 with the 16S Metagenomics workflow module (Life Technologies). Clustering of operational taxonomic units (OTUs) and taxonomic classification were performed based on two-step Basic Local Alignment Search Tool (BLAST) that maps to two separate reference libraries. In the first step, reads were aligned to the MicroSEQ® 16S Reference Library v2013.1 database (*Woo et al., 2003*). Subsequently, any unaligned reads subject to second alignment to the Greengenes v13.5 database (*DeSantis et al., 2006*). At least 10 of unique reads were valid and ≥90% for the alignment coverage needed between hit and query. Genus- and species-level identifications were accepted at ≥97% and ≥99% sequence identity, respectively (*Drancourt et al., 2000*). In case of any unusual microorganism, the sequence reads were reanalyzed by BLAST with the National Center for Biotechnology Information (NCBI) taxonomy database. Any OTUs represented by only one sequence (singletons) and those with fewer than 10 reads in the sample were excluded from further analysis. Alpha- (within sample) and beta- (among samples) diversities were calculated using Quantitative Insights Into Microbial Ecology (QIIME) software (*Caporaso et al., 2010*). Alpha diversity metrics, consisting of the Shannon diversity index, observed species and Chao1 index, were graphed using GraphPad Prism version 5.01 (GraphPad Software Inc., San Diego, CA, USA). The beta diversity metric was calculated according to Bray-Curtis dissimilarity index (*Bray & Curtis, 1957*) and displayed through a principal coordinates analysis (PCoA) plot. Heat map and hierarchical clustering tree were generated using R Studio (*RStudio Team, 2018*) employing R version 3.4.1 (*R Core Team, 2017*). Raw sequences were deposited in the NCBI Sequence Read Archive (SRA) accession number SRP158176.

## Statistical analysis

Statistical analysis was performed using IBM SPSS Statistics for Windows, version 19.0 software (SPSS Inc., Chicago, IL, USA). Demographic characteristics between two groups were compared using the chi-squared or Fisher's exact test. Mann–Whitney $U$ test for non-parametric data was used for bacterial abundance and diversity comparisons between two groups. $P$-values less than 0.05 were considered as statistically significant.

## RESULTS

### Participant characteristics

The median age and body mass index of the enrolled subjects were 39 years (ranging from 31 to 45) and 21.2 kg/m$^2$ (ranging from 16.0 to 35.8), respectively. Subjects fell into two groups: lactobacilli-dominated (LD) and non-lactobacilli dominated (NLD) groups (see below). No significant association between sociodemographic characteristics and VMB group (LD or NLD) was found, as shown in Table 1.

**Table 1 Sociodemographic and behavioral characteristics, and health habits of the studied subjects.**

| | Lactobacilli-dominated ($N = 14$) $n$ (%) | Non-lactobacilli dominated ($N = 11$) $n$ (%) | Total ($N = 24$) $n$ (%) |
|---|---|---|---|
| Median age (years) [range] | 40 [31–45] | 33 [27–45] | 39 [27–45] |
| Median BMI (kg/m$^2$) [range] | 21.1 [16.0–35.8] | 21.4 [19.3–30.4] | 21.2 [16.0–35.8] |
| Higher vocational or university education | 12 (85.7) | 10 (90.9) | 22 (88.0) |
| Currently married | 11 (78.6) | 10 (90.9) | 21 (84.0) |
| ≥1 child delivered | 10 (71.4) | 6 (54.5) | 16 (64) |
| Current smokers | 0 (0) | 0 (0) | 0 (0) |
| Income <400 USD per month | 4 (28.6) | 5 (45.5) | 9 (36.0) |
| Yogurt use ≥1 time per week | 13 (92.9) | 11 (100.0) | 24 (96.0) |
| Alcohol consumption ≥1 per week | 3 (21.4) | 3 (27.3) | 6 (24.0) |
| Exercise ≥1 time per week | 12 (85.7) | 8 (72.7) | 20 (80.0) |
| Sexual activity ≥1 time per week | 9 (64.3) | 9 (81.8) | 18 (72.0) |
| Sexual partner >1 time in previous year | 6 (42.9) | 1 (9.1) | 7 (28.0) |
| Regular ovulatory menstrual cycles in six months | 12 (85.7) | 10 (71.4) | 22 (88.0) |
| Current hormonal contraceptive use | 7 (50.0) | 2 (18.2) | 9 (36.0) |
| Vaginal douching done in previous month | 1 (7.1) | 1 (9.1) | 2 (8.0) |
| Abnormal discharge in previous month | 0 (0) | 2 (18.2) | 2 (8.0) |
| Female sterilization | 5 (35.7) | 4 (36.4) | 9 (36.0) |

**Notes.**
BMI, body mass index; USD, United States Dollar.
All variables were self-reported.
Statistically not significant for any factor. $P$-values were calculated using chi-squared or Fisher's exact analysis for assessment of association of frequency between groups and the Mann–Whitney $U$-Test for comparison of means.

## Analysis of VMB in the studied subjects

High-throughput sequencing data of all vaginal samples derived from Ion Torrent PGM yielded a total of 4,775,332 raw sequences with an average of 191,013 reads per sample (ranging from 33,969 to 382,574 reads). We discarded any reads shorter than 150 bp and those OTUs represented fewer than 10 times in the sample. This left a total of 4,189,265 valid reads with an average of 167,570 reads per sample (ranging from 28,825 to 333,480 reads) for analysis. The average length of included reads was 243 bp (ranging from 237 to 253 bp). A total of 3,553,612 mapped reads were clustered into OTUs and given taxonomic identification. Forty-one genera and 72 species were identified (Tables S1 and S2). VMB were clustered into two distinct groups: LD ($n = 14$) and NLD ($n = 11$). In the LD group, the average of relative abundance of *Lactobacillus* species is 90% and the relative abundance of this species among the LD individuals is ≥60% and could be further divided into two subgroups: one dominated by *L. iners* ($n = 10$) and the second subgroup ($n = 4$) was dominated by *L. crispatus*, *L. gasseri*, *L. jensenii* and *L. johnsonii*. The NLD group could similarly be divided into two subgroups: one dominated by *Gardnerella vaginalis* ($n = 3$) and the other ($n = 8$) dominated by anaerobic bacteria of a mixture of several species including *G. vaginalis, Atopobium vaginae* and *Pseudomonas stutzeri*. Figure 1 represents the hierarchical clustering tree (Fig. 1A) and the heat map (Fig. 1B) of bacterial taxa

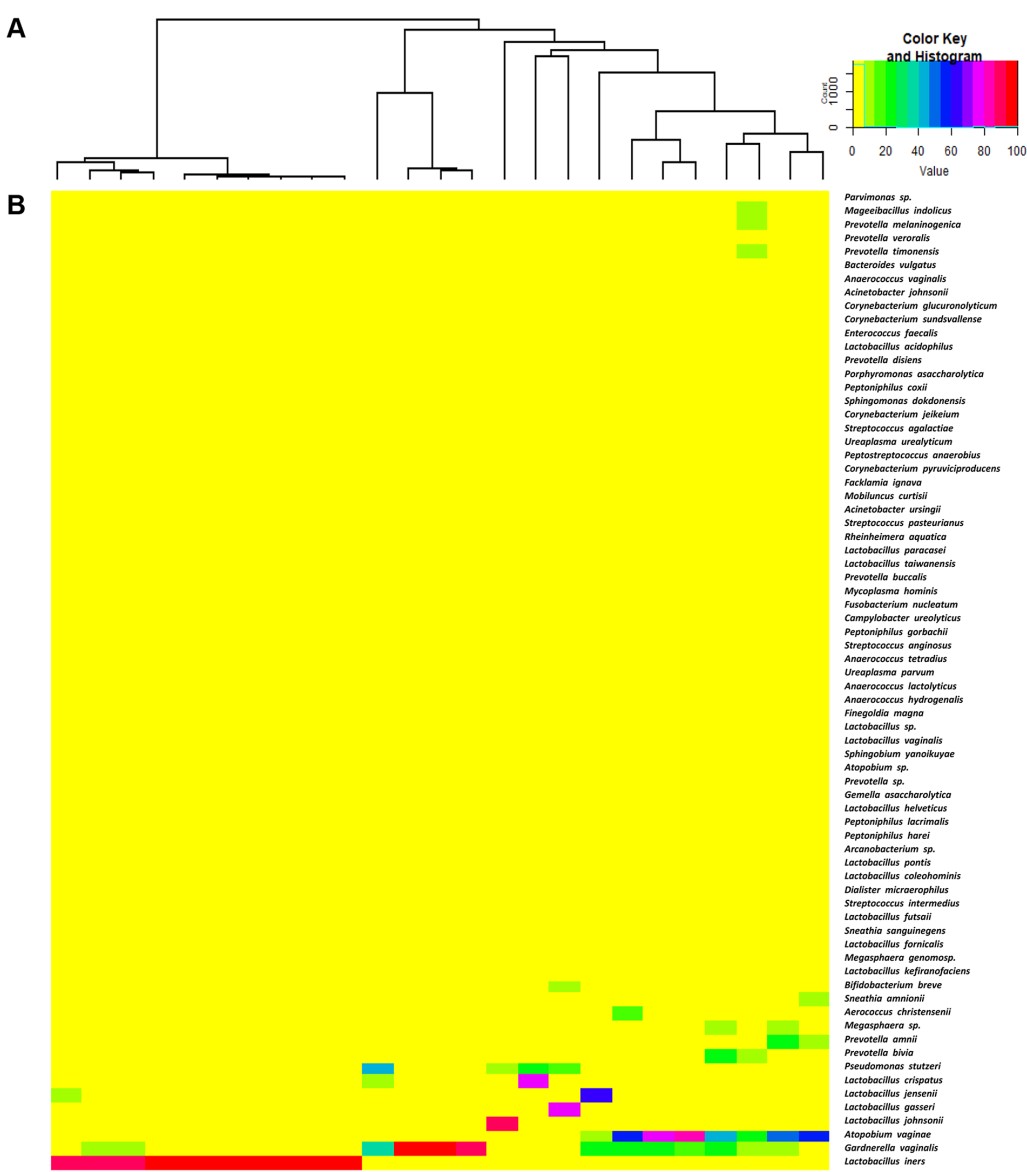

**Figure 1 The hierarchical clustering tree and the heat map of bacterial taxa from vaginal samples.** The hierarchical clustering of 25 vaginal microbiota was generated based on the bacterial abundances of OTUs (A). The heat map showing the relative abundance of the most abundant OTUs using the color key (B). The red bar and blue bar below the tree indicate the lactobacilli-dominated and non-lactobacilli dominated group, respectively.

identified from 25 vaginal samples. Table S3 indicates the mean percentage representation of each bacterial genus present in these two groups.

In this study, alpha-diversity for determination of bacterial diversity within individual women used three metrics: Shannon diversity index (estimated evenness and richness), observed species (observed bacterial richness) and Chao1 (estimated bacterial richness). The Shannon diversity index differed significantly between the two groups ($P < 0.001$)

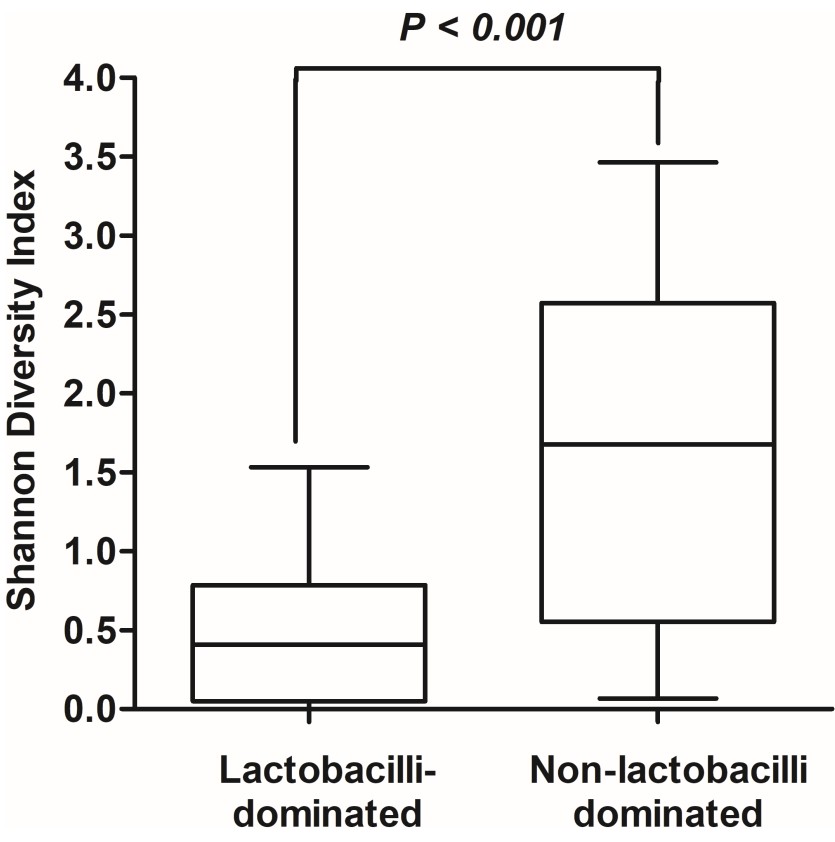

**Figure 2** **Shannon diversity index values in lactobacilli-dominated and non-lactobacilli dominated groups.** The Shannon diversity index values indicating diversity of bacterial taxa in both groups, as shown using Tukey's boxplots.

(Fig. 2). The observed and estimated numbers of OTUs in individuals in the NLD group were significantly higher than those in the LD group (both $P < 0.001$) (Fig. 3). All these results indicated that the bacterial communities in NLD individuals have more species diversity than those in LD individuals. In beta-diversity analysis, the PCoA plot based on taxonomic profiles showed that some distinct clustering appeared to overlap between two groups (Fig. 4). This result suggested that the VMB of both LD and NLD individuals did not tightly cluster by group. In addition, the relative distances based on the Bray-Curtis dissimilarity index showed significantly greater distances among LD individual samples than among NLD individual samples ($0.82 \pm 0.10$ versus $0.72 \pm 0.09$, $P = 0.021$). Since the distance measurement indicates the degree of similarity among samples (*Van de Wijgert & Jespers, 2016*), this result indicates that VMB in LD individuals resemble one another more closely than do those in NLD individuals.

The number of species unique to either LD or NLD, or common to both groups, were 14 (19.44%), 41 (56.95%) and 17 (23.61%), respectively (Fig. 5). The 17 common bacterial species found in both groups were *A . vaginae, Finegoldia magna, G . vaginalis, L. crispatus, L . gasseri, L . iners, L . johnsonii, L . kefiranofaciens, L . vaginalis, Megasphaera* spp., *Peptoniphilus harei, Peptostreptococcus anaerobius, Prevotella bivia, P . timonensis,*

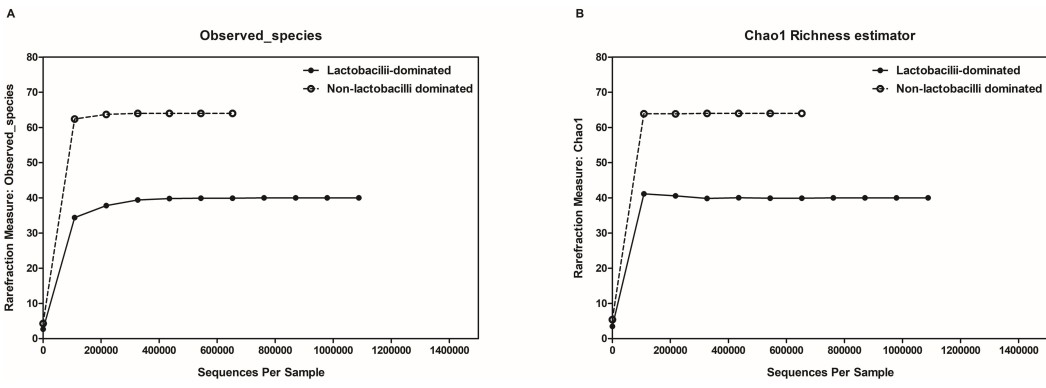

**Figure 3** **Rarefaction curves for observed species (OTUs) (A) and Chao1 (B).** Both were used to estimate detected bacterial richness in lactobacilli-dominated and non-lactobacilli dominated groups.

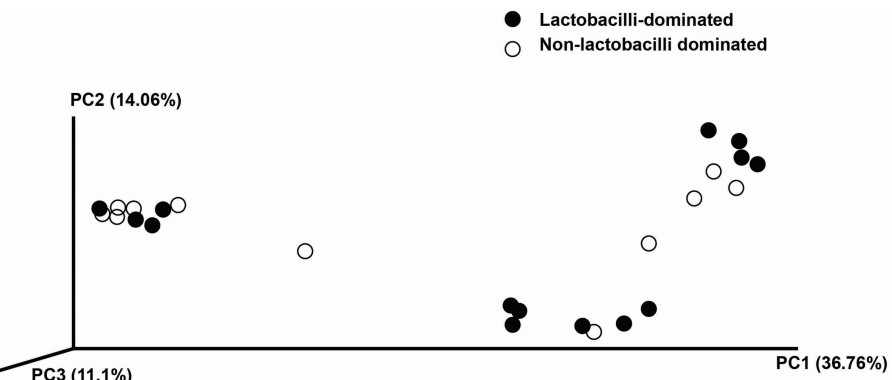

**Figure 4** **Principle coordinates analysis (PCoA) plot of Bray–Curtis dissimilarity indices among all samples.**

*Pseudomonas stutzeri*, *Streptococcus intermedius* and *Ureaplasma parvum*. Fig. 6 represents the most abundant bacterial species detected in the vaginal samples of LD and NLD groups. *G.vaginalis*, *A . vaginae* and *P. bivia* were significantly associated with the NLD group ($P < 0.05$) while *L. iners* was significantly associated with the LD group ($P < 0.05$). The bacterial species with abundance less than 1% are *F . magna*, *L. kefiranofaciens*, *L. vaginalis*, *Peptoniphilus harei*, *Peptostreptococcus anaerobius*, *S . intermedius* and *U. parvum*.

## DISCUSSION

The community composition of the VMB can vary greatly among women. However, although it can be absent entirely, *Lactobacillus* is likely to be the dominant, and sometimes only, genus in VMB (*Cherpes et al., 2008b*; *Linhares et al., 2011*; *Ma, Forney & Ravel, 2012*; *Petrova et al., 2015*; *Smith & Ravel, 2017*). We found the VMB of normal Thai women to fall into two groups. In the LD group, lactobacilli, and especially *L. iners*, can make up 90% of the bacterial cells present. In the NLD group, *G. vaginalis* followed by

**Lactobacilli-dominated**     **Non-lactobacilli dominated**

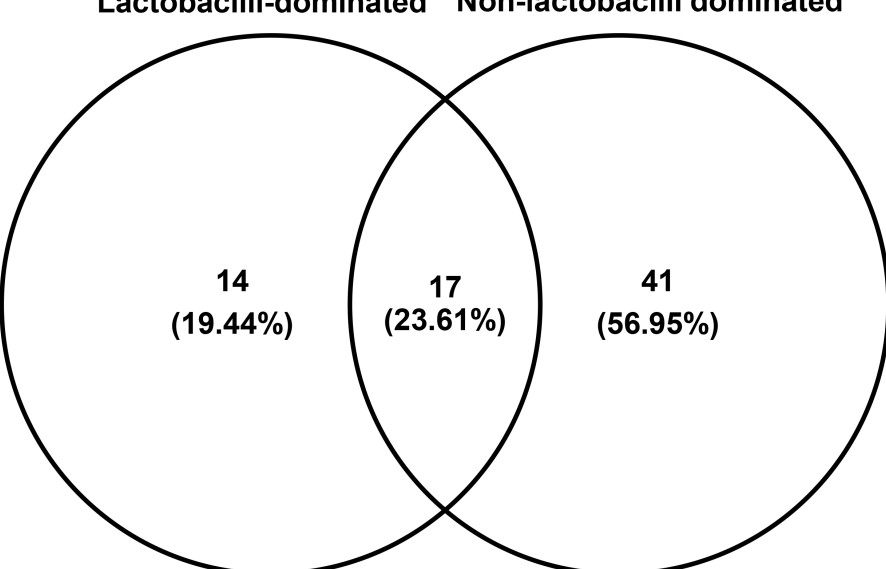

**Figure 5**   Venn diagram showing numbers of unique and shared OTUs (species) in lactobacilli-dominated and non-lactobacilli dominated groups.

*A. vaginae* and *Pseudomonas stutzeri* jointly constitute the majority of bacteria present. Previous studies found the difference on the composition of VMB among reproductive age women of different ethnicity (*Fettweis et al., 2014*; *Ravel et al., 2011*). Studied on VMB of asymptomatic North American women by *Ravel et al. (2011)* using Roche 454 sequencing (454 sequencing) of the V1–V2 regions of 16S rRNA gene found that *Lactobacillus*-dominated (80–90%) is the most common type of VMB in Asian and White women but less common (approx. 60%) in Hispanic and African American women. Furthermore, a *Lactobacillus*-dominated VMB was also found in normal Chinese and Korean women by using 454 sequencing of the V3 and V3–V5 regions, respectively (*Hong et al., 2016*; *Ling et al., 2010*). A high abundance of *Lactobacillus* species is generally considered as a good biomarker for a healthy vaginal ecosystem (*Petrova et al., 2015*). Lactobacilli promote a protective environment in the vagina by several mechanisms, such as production of hydrogen peroxide, bacteriocins or certain metabolites, that can inhibit the colonization and growth of various vaginal pathogens (*Aroutcheva et al., 2001*; *Eschenbach et al., 1989*).

    *Lactobacillus iners*, the dominant species of VMB in our study, is also the most common species in asymptomatic North American women with Asian ethnicity and in African American women by using 454 sequencing of the V1–V2 and V1–V3 regions of 16S rRNA genes, respectively (*Fettweis et al., 2014*; *Ravel et al., 2011*). Previous studies with different sequencing platforms of 16S rRNA gene, e.g., Illumina (V3–V4) and 454 (V1–V2 or V1–V3) sequencing, found other *Lactobacillus* species such as *L. crispatus*, *L. gasseri* and *L. jensenii* are commonly the dominant species in White or European women, but are less common in Asian, Black and Hispanic women (*Borgdorff et al., 2017*; *Fettweis et al., 2014*; *Ravel et al., 2011*). The difference in VMB composition likely have a genetic and
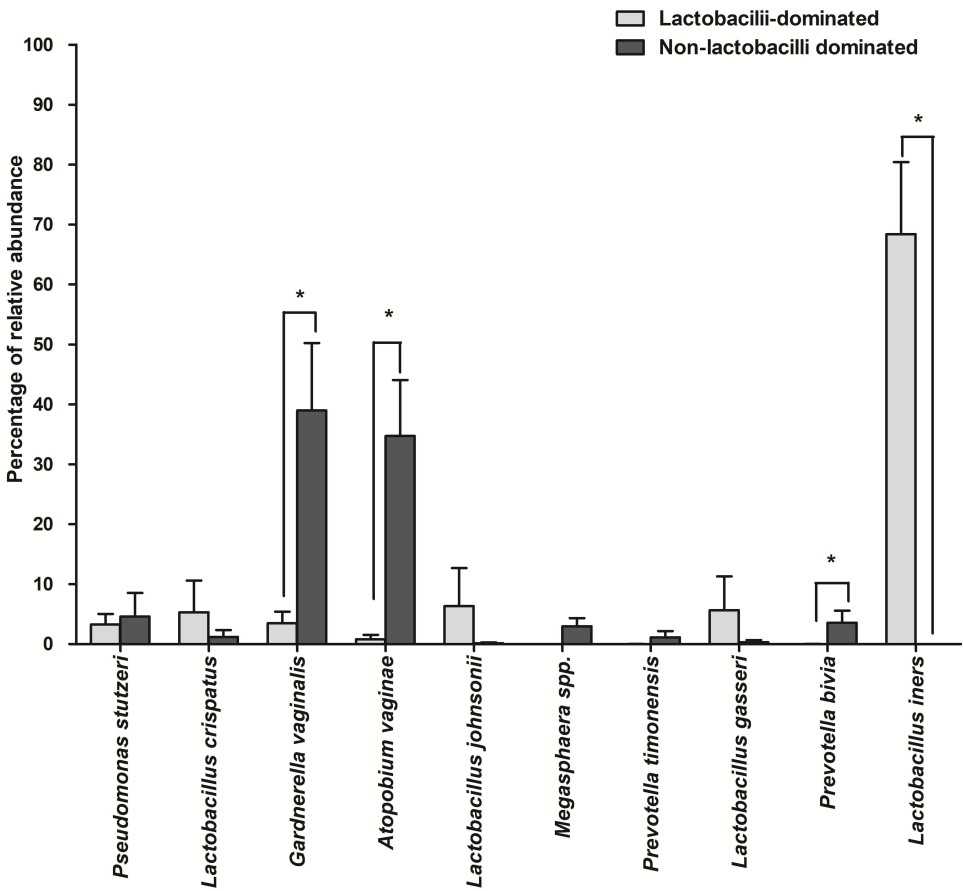

**Figure 6** **The most abundant bacterial species detected in vaginal samples in the lactobacilli-dominated and non-lactobacilli dominated groups.** Data are shown as mean with SEM. The bacterial species with abundance less than 1% are not shown. These include *F. magna, L. kefiranofaciens, L. vaginalis, Peptoniphilus harei, Peptostreptococcus. anaerobius, S. intermedius* and *U. parvum* . * = significant difference.

ethnic basis (*Green, Zarek & Catherino, 2015*). Nevertheless, a systematic review by *Van de Wijgert et al. (2014)* found that *L. iners* is present in most women, either healthy or with BV, while *L. crispatus* is commonly found only in healthy women. Longitudinal studies have found that women who have *L. iners*-dominated VMB at baseline are more likely to transition to a BV-associated VMB than women who have *L. crispatus*-dominated VMB. *Lactobacillus crispatus*-dominated VMB more often transitions to an *L. iners*-dominated or mixed lactobacilli VMB than to a BV-associated VMB (*Gajer et al., 2012*). The role of *L. iners* remains controversial. It appears that *L. iners* is well adapted to the vaginal niche, able to survive in a BV-like environment and often persisting after antibiotic treatment (*Huang et al., 2014*; *Van de Wijgert et al., 2014*). It may help to restore a lactobacilli-dominated microbiota during and after dysbiosis and/or after antibiotic treatment (*Van de Wijgert et al., 2014*). Thus, *L. iners*-dominated VMB might indicate a transitional condition between normal and abnormal. However, since our participants with *L. iners*-dominated VMB

show no symptom both before and after sample collection perhaps this is "normal" for Thai women. Other studies have proposed that genetic/ethnic differences in immune responses might make the vaginal mucosa more favorable for colonization by *L. iners* over *L. crispatu* s in African women (*Doh et al., 2004*; *Nguyen et al., 2004*; *Ryckman et al., 2008*). However, the exact mechanism leading to transitions in the microbiota are not known and might be affected by various factors, e.g., the hormonal changes, sexual intercourse, smoking, personal hygiene, antibiotic treatment, ethnicity, individual differences in immune status/response, vaginal epithelial cells and their secretions (*Petrova et al., 2015*; *Petrova et al., 2017*; *Van de Wijgert et al., 2014*).

Nearly half of the subjects (11 of the 25 subjects) was dominated by a suite of species, not by a single taxon. This type of VMB was previously found in 40.4%, 19.8% and 38.5% of asymptomatic Black or African women (*Ravel et al., 2011*), Asian women (*Ravel et al., 2011*), and women in Amsterdam (*Borgdorff et al., 2017*), respectively. Previous systematic reviews have revealed that some healthy women have this type of VMB (*Huang et al., 2014*; *Petrova et al., 2015*), which is dominated by members of the genera *Gardnerella*, *Atopobium*, *Prevotella*, *Corynebacterium*, *Anaerococcus*, *Peptoniphilus*, *Mobiluncus* and *Sneathia*. These species have evolved mechanisms to persist in a slightly alkaline environment and to adhere to the vaginal epithelial cells. It has been suggested that such a suite of bacteria is able to maintain the protective function of the vagina through lactic acid production to lower vaginal pH (*Gajer et al., 2012*). Lactic acid can be produced by either homolactic or heterolactic acid fermentation by certain facultatively or strictly anaerobic bacteria such as *Atopobium*, *Streptococcus*, *Staphylococcus*, *Megashara* and *Leptotrichia* (*Petrova et al., 2015*; *Zhou et al., 2004*). These bacteria might contribute to maintain the balance of the healthy vaginal ecosystem. Thus, the polymicrobial VMB seen in our and other studies indicated that this type of VMB can be present in normal healthy women (*Petrova et al., 2015*).

It should be noted that in the study of VMB using 16S rRNA gene sequencing there are difference in the usage of 16S rRNA region to generate reads and in the database for taxonomic classification among the studies. There is much speculation on which hypervariable region provides sufficient sequences diversity to identify the most bacteria accurately (*Liu et al., 2008*; *Schloss, Gevers & Westcott, 2011*). Many studies found that using different hypervariable regions of the gene provides variable results (*Barb et al., 2016*; *Chakravorty et al., 2007*; *Youssef et al., 2009*). The best solution would be to sequence the entire 16S rRNA gene; however, this approach is not possible with short read NGS platforms. In this study, we used Ion Torrent PGM platform which using multiple variable regions (V2–V4 and V6–V9 regions) to explore the VMB which would provide sufficient sequence to identify the most bacteria accurately. In addition, in the taxonomic classification in this study primarily performed by alignment to the MicroSEQ® 16S Reference Library v2013.1 database and the Greengenes v13.5 database as described in Materials and Methods above revealed some unusual microorganisms, i.e., *Olsenella umbonata* and *O. profusa*. The sequence reads of these two microorganisms were reanalyzed with NCBI database and found them to belong to the most common species, i.e., *A. vaginae* (% identity of *O. umbonata* and *A. vaginae* are 99 and 100, respectively, and the identity of both *O. profusa* and *A. vaginae* is 99%). For the other microorganisms, the classification using either

MicroSEQ® 16S Reference Library/Greengenes database or NCBI database found to give similar results.

Previous studies suggested that differences in bacterial composition in VMB might be associated with both intrinsic and extrinsic factors, e.g., ethnicity, sociodemography, environment, pregnancy, smoking status, sexual behavior, number of sexual partners, alcohol consumption and host genetic factors (*Borgdorff et al., 2017*; *Fettweis et al., 2014*; *Huang et al., 2014*). Five different VMB clusters were described among four different ethnic groups (White, Black, Hispanic and Asian women) in North America (*Ravel et al., 2011*). Two additional clusters, dominated by *G.vaginalis* or BV-associated bacteria (BVAB) type 1, were found in another study in African American women and women of European ancestry (*Fettweis et al., 2014*). Genomic markers have revealed that differences in VMB by ethnicity might be related to mitochondrial DNA (mtDNA) polymorphisms (*Blekhman et al., 2015*; *Green, Zarek & Catherino, 2015*; *Ma et al., 2014*). High estradiol during pregnancy has been reported to be associated with high levels of lactobacilli and low bacterial diversity (*Aagaard et al., 2012*; *Petricevic et al., 2012*), and with low incidence of BV and low prevalence of BVAB during pregnancy (*Romero et al., 2014*). Smoking is a risk factor of BV (*Cherpes et al., 2008a*; *Ryckman et al., 2009*) and smokers appear to have a lower proportion of vaginal *Lactobacillus* spp. than do non-smokers (*Brotman et al., 2014*). Sexual behaviors and number of sexual partners have been implicated the bacterial composition and diversity of the VMB (*Huang et al., 2014*; *Van de Wijgert et al., 2014*). Prior studies have shown that the bacterial diversity in women engaging in high-risk sexual behavior was increased along with the decline of lactobacilli (*Wessels et al., 2017*). Lastly, due to a small sample size in this study, the large sample size in future study is inevitably required to draw a more meaningful conclusion.

## CONCLUSION

Our study is the first to report the types of vaginal microbiota in normal Thai women using 16S rRNA gene sequence data obtained by NGS. Two major groups were recognized, i.e., lactobacilli-dominated and non-lactobacilli dominated groups. *Lactobacillus iners* is the dominant species of vaginal microbiota in the lactobacilli-dominated group while several species are abundant in vaginal microbiota in the non-lactobacilli dominated group. The information on VMB in Thai women is a starting point for further studying of factors involved in the development and maintenance of vaginal microbiota communities in vaginal health and disease. A better understanding of vaginal microbiota, including their interaction with external and internal factors, will assist in development of effective strategies to manage reproductive health of Thai women.

## ACKNOWLEDGEMENTS

We would like to acknowledge Professor David Blair for editing the manuscript via Publication Clinic KKU, Thailand.

### Funding

This study was supported by the Royal Golden Jubilee (RGJ)-Ph.D. program Grant (PHD/0133/2558) of the Thailand Research Fund (TRF), Thailand. The funders had no role in study design, data collection and analysis, decision to publish, or preparation of the manuscript.

### Grant Disclosures

The following grant information was disclosed by the authors:
Thailand Research Fund (TRF): PHD/0133/2558.

### Competing Interests

The authors declare there are no competing interests.

### Author Contributions

- Auttawit Sirichoat performed the experiments, analyzed the data, prepared figures and/or tables, authored or reviewed drafts of the paper, approved the final draft.
- Pranom Buppasiri conceived and designed the experiments, performed the experiments, authored or reviewed drafts of the paper, approved the final draft.
- Chulapan Engchanil performed the experiments, authored or reviewed drafts of the paper, approved the final draft.
- Wises Namwat and Kiatichai Faksri authored or reviewed drafts of the paper, approved the final draft.
- Nipaporn Sankuntaw analyzed the data, contributed reagents/materials/analysis tools, authored or reviewed drafts of the paper, approved the final draft.
- Ekawat Pasomsub and Wasun Chantratita contributed reagents/materials/analysis tools, authored or reviewed drafts of the paper, approved the final draft.
- Viraphong Lulitanond conceived and designed the experiments, analyzed the data, contributed reagents/materials/analysis tools, authored or reviewed drafts of the paper, approved the final draft.

### Human Ethics

The following information was supplied relating to ethical approvals (i.e., approving body and any reference numbers):
This study was approved by the Ethics Committee of Khon Kaen University (HE601017).

### DNA Deposition

The following information was supplied regarding the deposition of DNA sequences:
Raw sequences were deposited in the NCBI Sequence Read Archive (SRA) accession number SRP158176.

### Data Availability

The raw data are provided in the Supplemental Files.

## Supplemental Information

Supplemental information for this article can be found online at http://dx.doi.org/10.7717/peerj.5977#supplemental-information.

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
