# Peer review of "Characterization of vaginal microbiota in Thai women"

_PeerJ, doi:10.7717/peerj.5977_

## Round 0.1 · original submission · Minor Revisions

Ensure that adequate details are available in the methods, as per some reviewer comments. In particular, please indicate how depth of sequence was taken into account before calculating alpha or beta diversity (i.e., were the sequences rarefied?)

Statistics are often used in these types of exploratory studies in the absence of more sophisticated methods for explaining variation in microbiomes, however it is still good to either use non-parametric method or transform the data prior to using parametric methods like a t-test.

Reviewer 1 ·

Basic reporting

-Mainly professional English, a few areas need some attention.
-Additional references should be provided for certain statements, see General Comments
-Article well structured, and self-contained

Experimental design

-Yes, original research
-Research question relatively well defined, and fills a gap
-Methods and Analysis adequately described and performed.

Validity of the findings

-Novel findings, of interest to those in the field
-N is small
-Conclusions seem appropriate, however more speculative language is needed when referring to 'health' of the VMB (intro and discussion), see General Comments.

Additional comments

The manuscript entitled “Characterization of vaginal microbiota in normal Thai women” by Dr. Sirichoat et al., is a cross-sectional study describing the vaginal microbiota in Thai women (N=25), as there are no reports of the VMB in this ethnicity. Further, the vaginal microbiota have been found to vary by ethnicity, thus a study describing an unreported group is scientifically interesting. At present, this reviewer feels that the manuscript is suitable for publication in the PeerJ following minor revisions.

Minor comments:
-Consider removing the word ‘normal’ from the title… What is normal?
-The sample size is small and varied (ie. menstrual cycle stage not consistent, +/- contraceptive use, female sterilization included, etc.). It would be very interesting to add to this dataset and compare say women without female sterilization to those with. Consider adding additional samples, or creating groups that can be compared in future studies (ie. proliferative phase vs. secretory phase).
-In abstract, perhaps use ‘Conclusions’ rather than ‘Discussion’
-Abstract, discussion, change ‘are the’ to ‘is’
-Intro, line 46, change ‘especially’ to ‘in particular it’
-Add detail to Intro regarding how the VMB varies by ethinicy.
-Add additional references to lines 48-51.
-Be cautious in using the term ‘healthy’ (line 51). Just because a VMB is diverse does not necessarily appear to mean it is unhealthy (40% of Black women have diverse VMB and are asymptomatic… they are ‘healthy’).
-gasseri is incorrect in line 52.
-Most of the work on the effect of lactobacilli inhibiting pathoges has been done in vitro, please make this clearer (line 53-55).
- Change to ‘rRNA gene sequencing’ throughout manuscript
-Change strictly to ‘strict’ (line 63)
-Add references to line 62-66, BV
-NGS studies in Canadian and American women are available, please add the references to line 67
-Methods: More thorough description of inclusion/exclusion. Ie. ‘use of vaginal douche in the past’… how long?
-Results: change heading to ‘Participant Characteristics’
-Is 90% relative abundance of Lactobacillus the typical cut-point used in the literature? This reviewer typically sees >50%
-Add p value for trend in line 215.
-Statements about transition of VMB (ie line 252), this implies that all the iners dominant women in the study are abnormal… but perhaps this is ‘normal’ for Thai women. Please add a statement to address.

·

Basic reporting

There is insufficient methodological detail to replicate this analysis.

Raw sequencing data, and perhaps more importantly, OTU representatives (to allow readers to re-classify by other methods), do not appear to be provided.

Experimental design

My most significant concerns with this study are methodological:

- The software used for classification was the "16S Metagenomics workflow module" of software provided by Life Technologies. I'm guessing that this a description of the product (from a google search): https://www.thermofisher.com/content/dam/LifeTech/Documents/PDFs/Ion-16S-Metagenomics-Kit-Software-Application-Note.pdf. This method may in fact perform well in comparison to other methods, but because one of the implicit goals of the project was to compare the vaginal microbiome of Thai women to that of other human populations, the methods used pose a challenge: how can we compare these results to those of other studies given the difference in technology (Ion Torrent vs Illumina), primer design (multiple amplicons vs a more typical single-aplicon design), and classification approach (sequence identity to proprietary databases)? This is course not a problem unique to this study, but the divergence of methods here poses particular challenges. The authors should discuss the implications of selection of this combination of methods for the ability to compare with other studies.

- A number of unfamiliar organisms are described here: have all of the Lactobacillus species reported here been described in the vagina (or in humans for that matter) in other studies? Here's a complete list:

Lactobacillus_kefiranofaciens
Lactobacillus_futsaii
Lactobacillus_crispatus
Lactobacillus_sp.
Lactobacillus_taiwanensis
Lactobacillus_paracasei
Lactobacillus_coleohominis
Lactobacillus_gasseri
Lactobacillus_vaginalis
Lactobacillus_fornicalis
Lactobacillus_helveticus
Lactobacillus_johnsonii
Lactobacillus_acidophilus
Lactobacillus_iners
Lactobacillus_jensenii
Lactobacillus_pontis

Many of these are familiar, but some seem very unlikely: from a quick search, L kefiranofaciens, L futsaii, and L paracasei (for example; I stopped looking at this point) are environmental or agricultural organisms. Has Olsenella umbonata (mentioned in the abstract) been described in other studies of this environment? This species appears to have been first described based on isolates from pigs and sheep (https://doi.org/10.1099/ijs.0.022954-0) - has it been recovered from humans? Based on the examples above, the reader should be skeptical of the classification methods. It is my belief that the burden of evidence for the performance of proprietary software should be high given the difficulty of reproducing the analysis by other groups. Can the authors suggest an approach for verifying the validity of the classification results for organisms in this environment?

- The lack of correlation with sociodemographic and behavioral factors is not surprising given the small size of the cohort - I'd suggest that this study was probably underpowered for such a comparison. The absence of correlation can probably not be interpreted strongly.

- Similarly, it also is not clear to me that the size of the cohort is sufficient to draw meaningful conclusions about the similarity or differences between the microbiome of this population and others.

- The heatmap (Fig 1) appears to show absolute read counts rather than relative abundance - this is problematic if read depth varies among specimens.

- I am not a statistician, but one should weigh in on the method ("Independent t-tests for parametric data were used for bacterial abundance) used in Figure 6. Lots of concerns here, but in particular: 1) there is no mention of correction for multiple comparison; 2) there are well-known problems with induced correlation due to compositional data at work here that make this approach likely to be inappropriate, see (for example) https://www.ncbi.nlm.nih.gov/pmc/articles/PMC5695134/.

Validity of the findings

See above for comments on statistical inference and robustness of the data and analysis.

Additional comments

This manuscript presents bacterial community profiling data from a small (N=25) cohort of healthy Thai women. A combination of 16S rRNA variable regions were sequenced using an Ion Torrent protocol; classifications were performed by comparison with two reference databases using proprietary software. Lactobacillus-dominant and non-Lactobacillus-dominant community types were identified and contrasted. Sociodemographic and behavioral factors were also compared with community composition, none of which where found to be significantly associated. The manuscript is well-written, but I have significant concerns about the methods; see individual sections above.

---

## Round 0.2 · Minor Revisions

Thank you for revising your manuscript and addressing many of the reviewers' comments. The following is a list of outstanding issues that I feel need to be addressed before the manuscript can be considered for publication.

1. Reviewer 2 raised the concern that your interpretation of the results is in the context of Illumina sequencing of the 16S rRNA gene whereas you used Ion Torren PGM sequencing. Please point out this fact and reference studied of the vaginal microbiome that were done using the same sequencing technology. Please also mention that different variable regions of the 16S rRNA gene give variable results.

2. The sample size is too small for statistical analysis of the association of SDS with the vaginal microbiome and the results do not trend therefore, the section "Factors associated with VMB" should be removed.

3. Please take a close look at the methods and ensure that all data is correct and complete. For example the sentence on Ln 191 "We discarded any reads [...] represented fewer than 10 times in any given sample" implies that if an OTU was not present one sample at >10 reads it was discarded from all other samples.

4. Sequence data should be deposited in a public database and the accession number added prior to publication.

---

## Round 0.3 · Minor Revisions

You have addressed nearly all reviewer comments appropriately, however, you must account for sequencing depth between samples when analyzing them together. The depth of sequence varies by an order of magnitude making it inappropriate to compare absolute read counts between samples. For the heatmap in figure 1 and all analyses that involve comparing samples to one another please either use the relative abundance for OTUs or a rarefied dataset.

---

## Round 0.4 · accepted · Accept

Thank you for addressing the concerns

#